# Photoactivated nanomotors via aggregation induced emission for enhanced phototherapy

Shoupeng Cao[1,4], Jingxin Shao[1,4], Hanglong Wu[1], Shidong Song[1], Maria Teresa De Martino[1], Imke A. B. Pijpers[1], Heiner Friedrich [2], Loai K. E. A. Abdelmohsen [1✉], David S. Williams[3✉] & Jan C. M. van Hest [1✉]

Aggregation-induced emission (AIE) has, since its discovery, become a valuable tool in the field of nanoscience. AIEgenic molecules, which display highly stable fluorescence in an assembled state, have applications in various biomedical fields—including photodynamic therapy. Engineering structure-inherent, AIEgenic nanomaterials with motile properties is, however, still an unexplored frontier in the evolution of this potent technology. Here, we present phototactic/phototherapeutic nanomotors where biodegradable block copolymers decorated with AIE motifs can transduce radiant energy into motion and enhance thermophoretic motility driven by an asymmetric Au nanoshell. The hybrid nanomotors can harness two photon near-infrared radiation, triggering autonomous propulsion and simultaneous phototherapeutic generation of reactive oxygen species. The potential of these nanomotors to be applied in photodynamic therapy is demonstrated in vitro, where near-infrared light directed motion and reactive oxygen species induction synergistically enhance efficacy with a high level of spatial control.

[1] Bio-Organic Chemistry, Institute for Complex Molecular Systems, Eindhoven University of Technology, P.O. Box 513, MB Eindhoven, The Netherlands. [2] Center for Multiscale Electron Microscopy (CMEM) and Department of Chemical Engineering and Chemistry, Physical Chemistry, Institute for Complex Molecular Systems (ICMS), Eindhoven University of Technology, MB Eindhoven, The Netherlands. [3] Department of Chemistry, College of Science, Swansea University, Swansea, UK. [4] These authors contributed equally: Shoupeng Cao, Jingxin Shao. ✉email: l.k.e.a.abdelmohsen@tue.nl; d.s.williams@swansea.ac.uk; j.c.m.v.hest@tue.nl

Synthetic nano and micro-motors/swimmers capable of self-propulsion have received increasing attention due to their ability to harness non-Brownian motility for active cargo transport[1–5]. An array of nanoscopic motors have been developed with various modes of chemical and physical propulsion[6–10]. Significant limitations however still hamper translation of such nanomotor systems to biomedical applications, associated with their composition and mechanism of propulsion and reliance on (for example) high levels of chemical fuel, steep chemical gradients, non-biocompatible components, and environmental sensitivity/deactivation[5,11–13]. In order to become applicable in a biomedical context, it has become apparent that it is vital to create motile nanosystems that are able to employ non-chemical (e.g., radiative) energy for site-specific activity (i.e., toxicity), thereby integrating propulsion and therapeutic function[14–17]. To achieve this goal, intricately designed nanoparticles are needed, with physical features that can be incorporated in a robust way. Through functional synergy between integrated physical elements, particle design can be kept simple and performance tailored towards the biomedical application in mind.

Aggregation-induced emission (AIE) is a feature that clearly meets these criteria. AIE has been explored as a powerful tool for biomedical applications. First-generation AIEgenic molecules have demonstrated potential in photodynamic therapy (PDT), outperforming conventional photo-therapeutics that suffer from low loading/conjugation efficiencies, burst release, aggregation-induced quenching (ACQ), or photo-bleaching behavior[18–20]. Second-generation AIEgenic compounds offer increased performance in terms of two-photon near-infrared (TP-NIR) activation and improved reactive oxygen species (ROS) generation[21,22]. Coherent TP-NIR activation is a powerful tool for deep-tissue imaging and therapy, where it can highly selectively activate NIR-absorbing materials (spatial resolution of 1 $\mu m^3$) and those capable of two-photon absorption[23,24]. It is well-positioned to become a powerful therapeutic modality as and when compatible technologies are developed to exploit its unique capabilities[23–25]. AIE-genic moieties have been incorporated in a range of different particles, and recently polymer vesicles (polymersomes) featuring AIEgenic fluorescent properties have been presented[26–29]. However, as of yet, these polymersomes have not been employed to

realize advanced physical or biological functionality, let alone motion[26,28]. In order to harness the phototherapeutic potential of AIEgenic polymersomes toward enhanced motility, another structural element is required that is able to translate radiative energy into a physical driving force for propulsion.

A potential candidate to complement AIEgenic polymersomes, which has been extensively used in nanomedicine, are plasmonic, nano-sized gold shells[30]. NIR-activated gold nanostructures have been shown to undergo photothermal heating, which can be used in photothermal therapy (PTT) for precise ablation of cancer cells by locally elevated temperatures[16,31–34]. Recently, the photothermal properties of plasmonic gold nanostructures have been employed to drive thermal motion (thermophoresis)[16,35,36]. By coating nanoparticles with a hemispherical gold nanoshell, NIR-induced photothermal motility has enabled cell internalization through highly directional motion and percolation of the plasma membrane[31,37]. As a complementary structural element, the plasmonic absorbance of an Au nanoshell would act as an energy sink for AIE fluorescence that will facilitate hitherto unseen synergy to drive both propulsion and phototherapy.

Herein, we present polymersomes comprising biodegradable copolymers which are covalently modified with a high density (>60 wt%) of a second-generation aggregation-induced emission (AIE) functionality[21,38]. AIE polymersomes are then coated with an asymmetric gold nanoshell to yield hybrid architectures that are activated using TP-NIR radiation[21,31]. Combining the properties of both AIE and Au, the advantages of both classes of material are captured whilst creating a therapeutic nanotechnology with increased efficacy and enhanced cytotoxic response upon targeted NIR irradiation. However, such functional modalities do not simply act concurrently but also in unison— whereby the AIE capacity transduces TP-NIR irradiation and, through highly stable fluorescence, photothermally activates the Au layer to provide enhanced motile behavior to increase the therapeutic effect. In this way, a higher level synergy is achieved where structure-inherent AIE (capable of absorbing TP-NIR for fluorescence and generation of reactive oxygen species) enhances the performance of an Au motor through transduction of radiant into plasmonic energy and subsequent thermophoresis (Fig. 1). These AIE-transduced nanomotors are demonstrated to be a

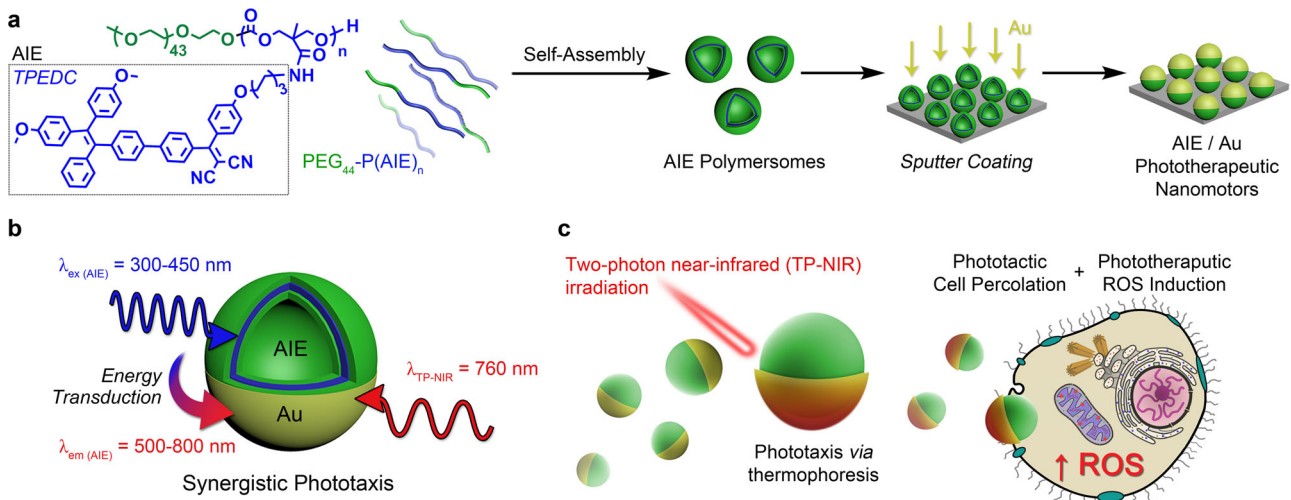

**Fig. 1 Design of synergistic AIE-transduced phototherapeutic nanomotors. a** Biodegradable copolymers, comprising AIEgenic TPEDC moieties, were designed for self-assembly into polymersomes, which were subsequently coated with gold to give hybrid AIE/Au nanomotors. **b** Synergistic motion underpins nanomotor performance where plasmonic Au is directly activated using two-photon near-infrared (TP-NIR, 760 nm) and indirectly activated via energy transduction from the AIE polymersome (excited by TP-NIR at 380 nm) resulting in enhanced phototaxis. **c** TP-NIR activation of nanomotors triggers dual behavior: enhancing cellular interactions and uptake (via percolation) alongside phototherapeutic ROS generation for highly localized cell toxicity.

powerful tool for site-specific therapy, triggered by an externally controlled physical stimulus (TP-NIR) without the need for chemical signaling.

## Results and discussion

**Molecular programming and engineering of AIE-polymersomes.** To realize the formation of biocompatible nanoscale motors with an innate susceptibility to TP-NIR we combined AIEgenic chemistry with the controlled self-assembly of biodegradable block copolymers to achieve AIE-modified polymersomes[26,39–41]. Nanomotors are particularly challenging to engineer due to the combined necessary control of structural asymmetry (related to the mechanism of propulsion) and nanoscopic dimensions[8,42]. A number of strategies have been published for this, including morphological engineering and multi-step assembly of patchy (asymmetric) nanostructures[43–47]. Utilizing the chemical versatility of poly(trimethylene carbonate) (PTMC) copolymers, numerous biofunctional nanosystems have been reported[48]. We prepared functionalized PTMC copolymers using pentafluorophenyl derivatives of TMC to generate well-defined polymers that could be modified (post polymerization) with an amine-containing molecule for the incorporation of functional moieties[49]. Such polymers have been shown to form spherical polymersomes, which can readily be implemented in biomedical research[50,51].

To engineer AIE polymersomes with functional capacity towards phototherapeutic generation of reactive oxygen species (ROS) and an increased two-photon absorption cross-section, we synthesized a "second generation" AIEgenic compound comprising both tetraphenylethylene and dicyanovinyl moieties (TPEDC) as developed by Liu et al. (Fig. 1a, Supplementary Figs. 1–13)[52]. AIE provides a powerful tool for the fabrication of light-activated nanoparticles, capable of enhanced phototherapy through the production of ROS[21]. In general, AIE molecules are formulated as nano-aggregates, however, there is an increasing ambition to integrate such functionality within more defined morphologies such as polymersomes[26,53]. To this end, we synthesized a range of well-defined amphiphilic poly(ethylene glycol) (PEG)-PTMC(TPEDC) copolymers (PEG$_{44}$-P(AIE)$_n$, Đ ≈ 1.1) to explore the effect of increasing steric bulk upon self-assembly in order to achieve functionally-dense, morphologically discrete AIE polymersomes (Supplementary Figs. 14–24 and Supplementary Table 1).

Having synthesized a range of PEG$_{44}$-P(AIE)$_n$ copolymers (where $n$ = 5, 8, 14, or 22), we investigated the effect of molecular composition upon self-assembly using a dropwise solvent switch process from THF to water (at 50 vol%) followed by dialysis purification (Supplementary Fig. 25)[54]. PEG$_{44}$-P(AIE)$_{5/8/14}$ copolymers were compatible with this process, assembling into nanoparticles of around 300–500 nm in size with low poly-dispersity (PDI ≤ 0.1) (Fig. 2a/b and Supplementary Figs. 26, 27 and Supplementary Table 2). Using transmission electron microscopy (TEM) and cryo-TEM the vesicular morphologies of these systems were readily identifiable with membrane thickness values commensurate with the length of the P(AIE) block (ranging from ca. 8–14 nm, Supplementary Table 2). Longer PEG$_{44}$-P(AIE)$_{22}$ copolymers were assembled using a fast precipitation method to avoid formation of visible aggregates and this approach yielded micelles/polymer nanoaggregates of ca. 70 nm (Supplementary Fig. 28). Due to the balance between increased density of AIEgenic groups in the polymersomal membrane and overall size, we chose to utilize PEG$_{44}$-P(AIE)$_{14}$ for the formation of AIE polymersomes, which were then applied throughout the rest of this work. Fluorogenic properties of AIE polymersomes were assessed by monitoring the transition from organic solution to the assembled state where strong emission

($\lambda_{ex}$ = 373 nm/$\lambda_{em}$ = 617 nm) was detected after addition of only 20% water (Fig. 2c/Supplementary Fig. 29). Strong AIE fluorescent properties are a powerful tool for both imaging of nanoparticles and phototherapeutic ROS induction[19,21].

**Fabrication of AIE-polymersome nanomotors and characterizations.** Having successfully generated tier 1 of our dual-functional nanosystem, the second functional element was introduced (Fig. 1a/b)[55]. NIR-triggered, photo-induced motility (phototacticity) was achieved through the deposition of a hemispherical gold nanoshell onto the AIE polymersomes. This was conducted using a recently published process and confirmed using electron microscopy and Energy Dispersive X-Ray (EDX) analysis (Fig. 2d, e/Supplementary Figs. 31–37/Supplementary Movies 1 and 2)[56]. To confirm the integrity and vesicular nature of AIE/Au nanomotors after drying and Au sputtering, cryo-TEM (Supplementary Figs. 34, 35, and 37) and cryo-electron tomography (cryo-ET, Supplementary Fig. 36, Supplementary Movies 1 and 2) were carried out to investigate the morphology changes of polymersomes and the distribution of coated Au. Our results show that the coated Au nanoshell was only distributed outside the AIE/Au nanomotor membrane (Supplementary Figs. 35–37), and broken nanomotors were barely observed. Importantly, the membrane structure appeared to be intact during the whole process, which can be clearly seen from different cross-sectional planes in cryo-ET reconstruction results (Supplementary Fig. 36). Although the shape of some AIE/Au nanomotors was slightly deformed due to the drying effects, we believe the vesicular shape can be recovered in the subsequent hydration, which was already confirmed by re-hydration of the dried AIE polymersomes (Supplementary Fig. 34). Furthermore, from the electron microscopy and light scattering results, the Au coating did not cause particle aggregation, with sizes maintained at around 400 nm (Fig. 2f/Supplementary Fig. 30), nor did it interfere with the fluorescent character of the underlying AIE bilayer, although absorbance increased across the visible-NIR range (Supplementary Fig. 39). With both functional elements (AIE/Au) successfully amalgamated through a facile assembly process, a TP-NIR activated synergistic phototactic/phototherapeutic nanomotor was prepared for further testing (Fig. 1b/c). The structural integrity of the resulting nanomotors, based upon a polymersome chassis, was confirmed by their ability to encapsulate and retain hydrophobic and hydrophilic cargo during the gold coating process (Supplementary Fig. 38).

**Motion analysis of the polymersome nanomotors.** Owing to the asymmetric structure of AIE/Au polymersomes, self-propulsion was achieved through light (NIR)-activated plasmonic heating, localized at the Au surface (Fig. 3a). In line with the literature, plasmonic heating increases with both time and irradiation intensity, therefore, it was necessary to consider both parameters during subsequent analysis[31–33,36]. Motion studies were conducted in PBS buffer using confocal laser scanning microscopy (TP-CLSM) equipped with a two-photon laser (TP-NIR, $\lambda_{ex}$ = 760 nm) to activate, observe and record particle movement. Fluorescent properties of AIE nanomotors enabled real-time tracking of movement (Fig. 3b/Supplementary Fig. 39)[19,57]. Upon TP-NIR irradiation, nanomotors underwent autonomous motion, with velocities ($V_{xy}$; measured in the xy plane) proportional to the intensity of light (Fig. 3c/Supplementary Figs. 40, 41, Supplementary Movies 3–5). In agreement with the extant literature, we posit that positive thermophoresis is the driving force propelling asymmetric, Au-coated polymersomes via plasmonic heating under TP-NIR[35,36]. Analysis of the XY trajectories indicated a transition of motion from diffusive (Brownian) to propulsive

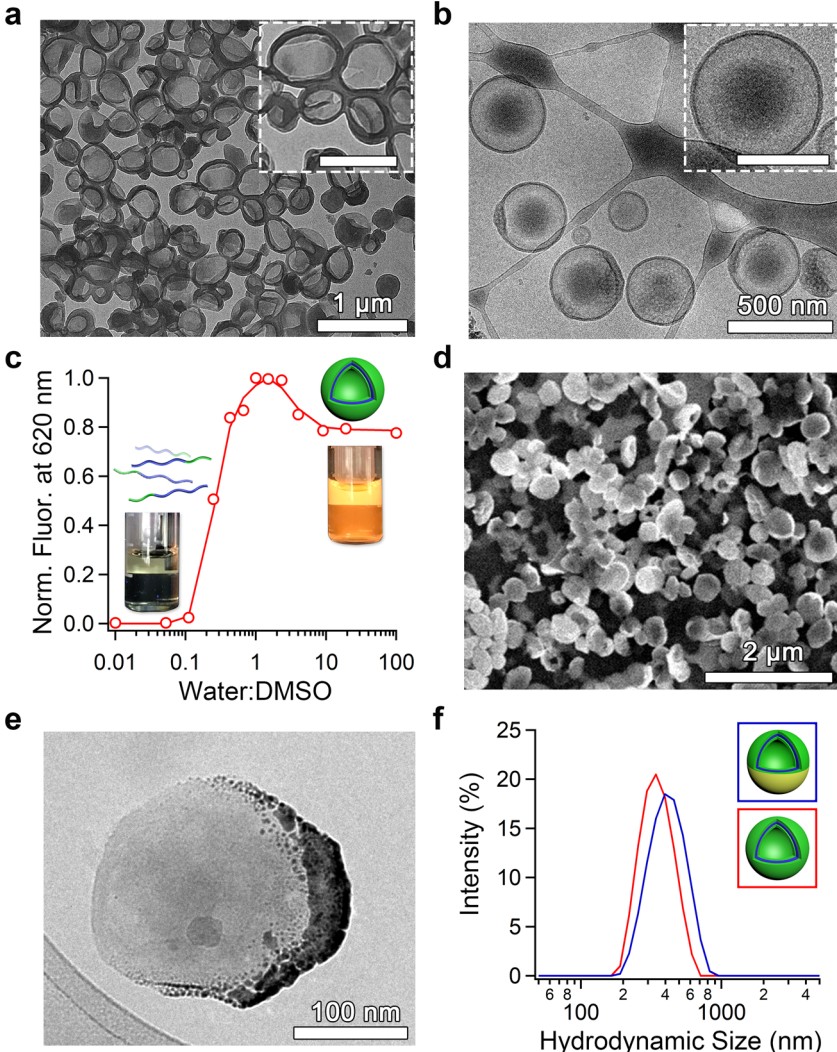

**Fig. 2 Physical characterization of AIE polymersomes and gold coating. a** Transmission electron microscope (TEM) (scale bar inset: 500 nm) and **b** cryogenic TEM images of AIE polymersomes (scale bar inset: 200 nm). **c** Aggregation-induced fluorescence of AIE copolymers arising due to their self-assembly at increasing water:DMSO ratios ($\lambda_{ex}$ = 373 nm/$\lambda_{em}$ = 617 nm). **d** Scanning electron microscopy (SEM) image of uniform gold-coated AIE nanomotors. **e** Cryo-TEM image of Au/AIE hybrid nanomotors with dark patches indicative of Au coating as confirmed by EDX analysis (Supplementary Fig. 33). **f** Particle size distribution using dynamic light scattering (DLS) of AIE polymersomes before and after gold coating.

under increased laser power as evident from the increasingly directional shape of the motion trajectories (Fig. 3b, c/Supplementary Figs. 41 and 42). Without laser power there was agreement between the theoretical values for the translational diffusion coefficient ($D_T$) of 400 nm nanoparticles (1.1 µm² s⁻¹) and the values obtained of 0.85 and 0.93 µm² s⁻¹ for AIE/Au and Au-only control non-AIE nanomotors (with sizes between 300 and 500 nm). Autonomous motion was investigated by correlating the mean-squared displacement (MSD) curves, and their slopes, with laser power. Using the diffusiophoretic model of Golestanian et al. motile particles that are being actively propelled should display a non-linear fitting at $\Delta t < \tau_r$ (where $\tau_r$ is the time required for a particle to make a complete rotation around its axis)[58,59]. However, the MSDs of hybrid nanomotors displayed a parabolic profile (indicative of autonomous propulsion) at longer timescales where $\Delta t > \tau_r$ (according to the Debye–Stokes–Einstein relation $\tau_r \approx 0.05$ s for such nanoparticles) for all laser powers used (Fig. 3c). To ensure that bulk photothermal heating (a property synonymous with plasmonic gold particles) was not responsible for the observed behavior, symmetric AIE-polymersomes and Au-coated

nanoparticles (160 nm NanoXact™ Gold Nanoshells, nano-Composix) were studied, and only enhanced Brownian diffusion (as opposed to propulsion) was observed under a standard NIR irradiation (Supplementary Figs. 49 and 50, Supplementary Movies 10–13). From our data, it was clear that the distinctly propulsive (also referred to as directional or ballistic) motion arising from AIE/Au nanomotors (and Au-only control particles) is an unexpected and unique property. In terms of performance, the velocities achieved by AIE/Au nanomotors were up to 45% higher than control Au-coated polymersomes, which did not contain structural AIE moieties (Fig. 3d/Supplementary Figs. 42 and 43). This observation points to the functional synergy between the AIE-rich polymeric framework and the Au coating, whereby the efficiency with which nanomotors (as a whole) were able to harness TP-NIR irradiation for propulsion was significantly improved as compared to non-functional AIE systems. Furthermore, the clearly observed propulsive behavior seems to challenge the theoretical limits surrounding motility due, in part, to the limits of current understanding associated with the exact mechanism of such nano-motors[60,61].

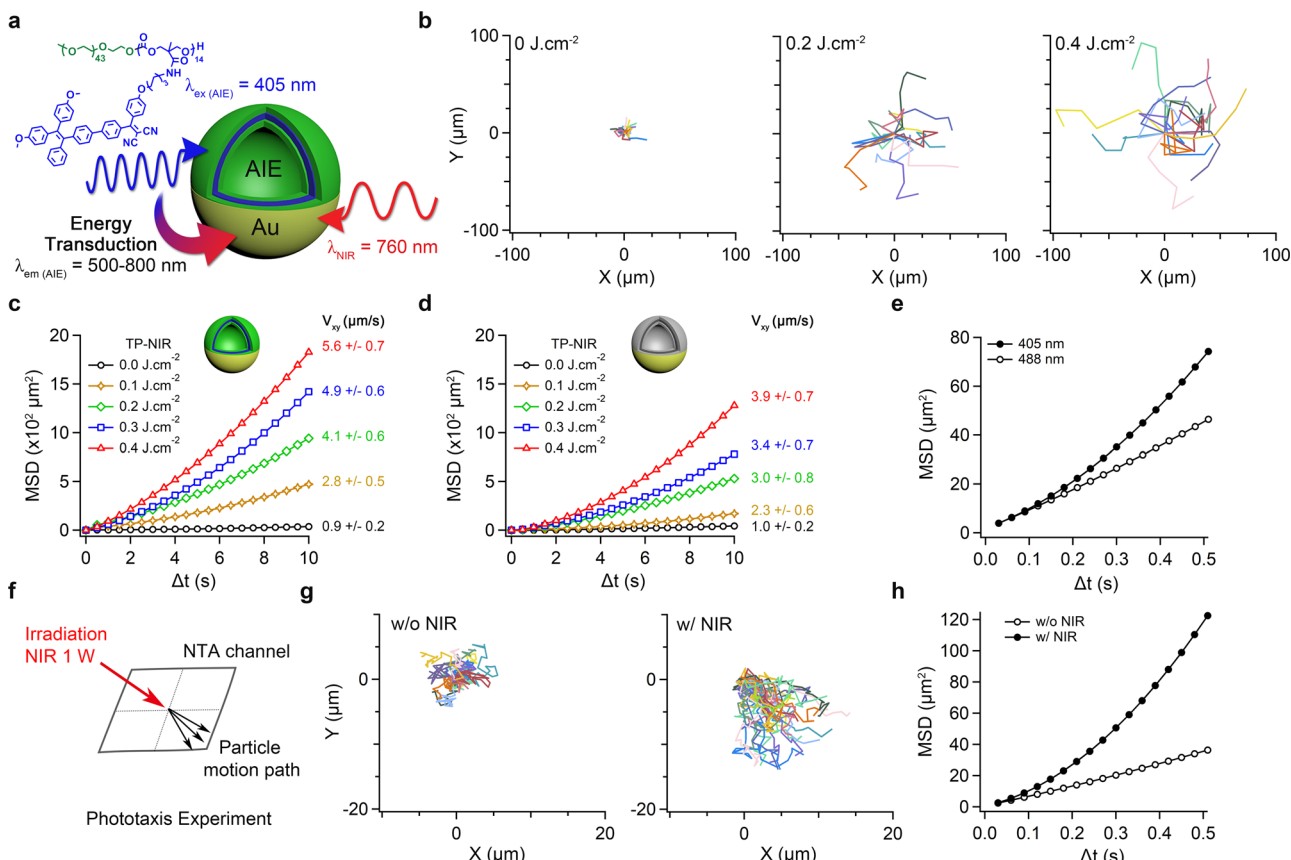

**Fig. 3 Synergistic phototaxis of AIE/Au nanomotors. a** AIE polymersomes transduce radiant energy into motion via the hemispherical Au nanoshell to synergistically enhance performance. **b** Particle trajectories of nanomotors measured using two-photon confocal laser scanning microscopy (TP-CLSM) at different laser powers. **c** Mean squared displacement (MSD) data for nanomotors at 760 nm (TP-NIR) and accompanying velocities derived from motion in the xy plane ($V_{xy}$) at varying laser power, alongside (**d**) MSD data for an Au-coated, non-AIE polymersome control (gray, cf. Supplementary Fig. 42). Brownian diffusion (random walk) and propulsive (directional) motion are identified by linear or non-linear MSD data, respectively. **e** MSD data (measured using Nanoparticle Tracking Analysis, NTA) for nanomotors comparing AIE-activated enhanced diffusion at 405 nm and Brownian diffusion at 488 nm. **f** Experimental setup to measure directional phototaxis using an NTA channel. **g** Directional particle trajectories of nanomotors under directional laser illumination (NIR, 660 nm) from the upper left and (**h**) accompanying MSD data with and without NIR illumination. Error bars depict the Standard Deviation (SD) from analysis by confocal tracking.

In terms of a mechanistic justification for the observed ballistic motion of Au-based nanomotors, we can speculate that highly localized heating (and the subsequent formation of asymmetric thermal gradients around our nanomotors) may give rise to non-classical behaviors overcoming the aforementioned theoretical limitations associated with rotational diffusion ($\tau_r$). However, further discussion would not be appropriate before further theoretical and experimental work has been undertaken—hopefully stimulated and supported by the present findings. In terms of performance, the enhanced behavior and superior velocities observed for hybrid AIE/Au nanomotors are postulated in Fig. 3a whereby structure-inherent AIE moieties, optimized for two-photon absorbance, bolster the energy efficiency of the system by transducing radiant energy into the plasmonic Au shell; enhancing thermophoresis and thereby increasing the propulsive effect. To prove this hypothesis, we used nanoparticle tracking analysis (NTA) to selectively excite AIE/Au nanomotors at 405 and 488 nm (Fig. 3e). As anticipated, non-Brownian (propulsive) motion was observed for AIE nanomotors only when using 405 nm irradiation (not at 488 nm) due to the selective excitation of AIE moieties at $\lambda_{AIE} < 450$ nm (Fig. 3e/Supplementary Fig. 44). In contrast, both sets of controls (un-coated AIE polymersomes and non-AIE nanomotors) only showed Brownian behavior using 405 or 488 nm excitation (Supplementary Figs. 45 and 46). This data

unequivocally established the ability of the polymeric AIE framework to transfer energy to the Au shell with sufficient efficacy to overcome rapid $\tau_r$ (owing to their nano-size), enabling dynamic motility under low-level activation. Such energization of plasmonic Au by the AIE polymer accounted for the increased velocity of the nanomotors. Further examination of particle trajectories revealed that the nanomotors (in response to standard NIR irradiation) tended towards directional motion away from the light source, which is known as negative phototaxis (Fig. 3f–h/ Supplementary Fig. 47, Supplementary Movies 6 and 7). Similar behavior has been observed for motile microswimmers that align along a diminishing intensity gradient[62], akin to the present scenario where laser light is reduced as it passes through the sample volume due to colloidal scattering. Such negative phototaxis can be harnessed to focus the delivery of such nanomotors and aid their penetration into biological tissue, which can be used to drive therapeutic modalities deeper into the target site to improve treatment efficacy[17,63–66].

**Enhanced photo-therapeutics with the AIE-polymersome nanomotors.** Having validated the synergistic performance of AIE/Au nanomotors in terms of motility, their potential for biomedical application, in particular, photodynamic therapy

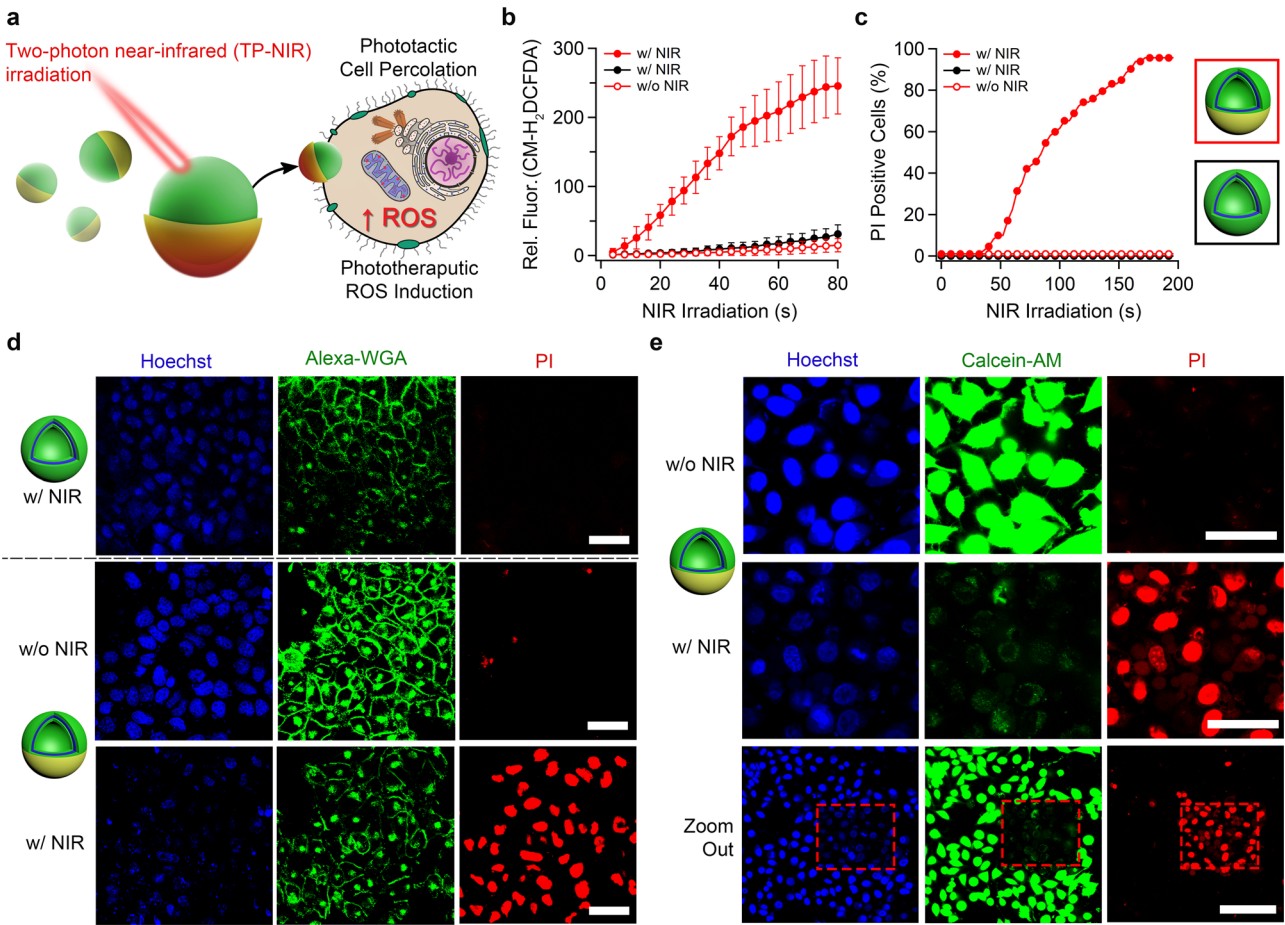

**Fig. 4 Biological performance of phototherapeutic nanomotors on cancer cells. a** Under TP-NIR activation, AIE/Au nanomotors undergo both propulsive motion (resulting in cell percolation) and phototherapeutic generation of reactive oxygen species (ROS) that synergistically combine to give potent, directed toxicity. **b** ROS induction, measured by the increasing fluorescence of pro-fluorescent ROS probe (CM-H₂DCFDA), and **c** apoptosis, measured according the proportion of HeLa cells that become permeable to propidium iodide (PI), as a function of total TP-NIR laser irradiation time (each sequential scan correlates to a 4 s pulse followed by imaging) comparing AIE polymersomes (black) with Au-coated nanomotors (red), with and without irradiation. **d** Confocal images of cell apoptosis as quantified in (**c**) after 200 s (with or without) TP-NIR irradiation where nanomotors outperform AIE polymersomes (nucleus: Hoechst, blue/plasma membrane: Alexa-WGA, green/apoptotic cells: PI, red). **e** Confocal images showing highly selective cell apoptosis using nanomotors with or without 200 s TP-NIR irradiation (nucleus: Hoechst, blue/viable cells: calcein-AM, green/apoptotic cells: PI, red). All scale bars = 50 μm. Error bars depict the Standard Deviation (SD) from analysis by confocal micrographs.

(PDT) was explored through a number of cell assays (Fig. 4a). The synergistic combination of NIR-induced self-propulsion and AIE-mediated generation of ROS was anticipated to yield highly effective and localized cell toxicity, providing a powerful tool for PDT. Firstly, the motion of the AIE/Au motor in cell medium was confirmed in response to a standard NIR light (Supplementary Fig. 48, Supplementary Movies 8 and 9). Experiments were conducted to study the interactions of nanomotors and control samples with cancer cells (HeLa). AIE polymersomes showed low level, passive uptake by cells over 24 h, although little signal was evident after 2 or 6 h (Supplementary Fig. 51). Au-coated AIE polymersomes had relatively low impact upon cell viability, with viability ≈ 70% at 200 μg mL⁻¹; compared to ≈ 85% for un-coated AIE polymersomes (Supplementary Fig. 52). TP-NIR irradiation of cells using a two-photon confocal laser scanning microscope (TP-CLSM) had no detrimental effect on cells even after 200 s total irradiation time (Supplementary Fig. 53). Furthermore, cells treated with AIE polymersomes (25 μg ml⁻¹) followed by TP-NIR irradiation did not show any significant toxicity (Supplementary Fig. 54). Cells also appeared relatively healthy when treated with AIE/Au nanomotors (25 μg ml⁻¹) without activation using NIR irradiation (Supplementary Fig. 55). These important control

experiments provide a backdrop to the stark behavior of AIE/Au nanomotors under TP-NIR irradiation.

Following treatment with AIE/Au nanomotors, plasma membrane integrity was monitored during 200 s of TP-NIR irradiation, resulting in significant disruption of the cell wall (Supplementary Fig. 56, Supplementary Movies 14 and 15). This disruptive behavior was likely due to the effect of highly propulsive nanomotors that caused localized percolation—a desirable outcome for photo-targeted therapy—and a pathway for enhanced incorporation into cells to improve the effect of photo-triggered ROS generation from the AIE polymer[31]. To assess the phototherapeutic aspects of nanomotors, intracellular ROS induction was monitored using the fluorogenic probe 2′,7′-dichlorodihydrofluorescein diacetate (CM-H₂DCFDA, Fig. 4b). In control experiments, AIE polymersomes induced a small degree of ROS in cells after 200 s TP-NIR irradiation (Supplementary Fig. 57). This low-level ROS induction from the small fraction of AIE polymersomes passively uptaken by cells, did not significantly impact cell viability even after 280 s of irradiation (Supplementary Fig. 58). In contrast, cells treated with AIE/Au nanomotors, and subsequently irradiated, showed high levels of ROS after only 48 s irradiation (Supplementary Fig. 59,

Supplementary Movie 16). Concurrently, enhanced accumulation of nanomotors within cells was detected using 3D confocal scanning (Supplementary Fig. 60) and the plasma membrane underwent catastrophic failure as indicated by high levels of propidium iodide (PI) sequestration (indicative of cell apoptosis) after only 80 s irradiation–in stark contrast to control studies (Fig. 4c, d/Supplementary Figs. 61 and 62). Confocal imaging of cells treated with nanomotors and a lysotracker stain did not yield any co-localization after irradiation, which suggested a trans-membrane trajectory of such particles, likely due to their high velocity and (relatively) small size (Supplementary Fig. 63).

To confirm that the sequestration of PI was associated with a loss of cellular activity, and an indicator of rapid-onset cell death, experiments were repeated using a "live" stain (calcein-AM, Supplementary Fig. 64, Supplementary Movies 17 and 18). Indeed, critical loss of cellular activity (indicated by diminished calcein fluorescence) after 40–80 s irradiation confirms previous assumptions that TP-NIR activated nanomotors are capable of highly controlled phototherapy. To showcase the level of spatial control provided by AIE/Au nanomotor technology, and to avoid experimental ambiguity, irradiated (and thereby terminated) cells were clearly identifiable alongside vicinal living cells, which would have been exposed to "dormant" particles without TP-NIR activation (Fig. 4e/Supplementary Fig. 65). The therapeutic implications of this technology are, therefore, that localized administration of such dual phototactic/phototherapeutic nano-motors can be combined with focused two-photon NIR irradiation to trigger highly specific and controllable ablation of diseased tissue. Dose-dependence studies of cells treated with nanomotors highlighted that cellular accumulation, PI sequestra-tion, and cellular activity were proportionately reduced at lower concentrations (from 25 down to 2.5, 0.5, and 0.25 $\mu g\,mL^{-1}$, Supplementary Figs. 66 and 67). This synergistic behavior between phototactic Au and the phototherapeutic AIE framework provides these nanomotors with a highly effective mode of directed cellular therapy in response to TP-NIR irradiation. This unique combination facilitates enhanced penetration of hybrid nanomotors into cells and tissues, where they can be activated to generate highly localized ROS to selectively treat disease[64]. Without AIE functionality, control particles (polymersomes coated with an asymmetric Au nanoshell) did not induce any detectable ROS within cells (Supplementary Figs. 68 and 69), and did not impact cellular viability under TP-NIR irradiation (Supplementary Fig. 70). As proof-of-principle, we highlighted the ability of AIE/Au nanomotors to undergo enhanced penetration and induce cytotoxicity in a 3D cell model comprising multi-cellular spheroids (Supplementary Fig. 71). Significantly, toxicity on the crown of the 3D tumor spheroids was evident when nanomotors were activated with TP-NIR irradiation, adding further weight to our findings, supporting the potential of this dual-functional nanomotor technology as a controllable phototherapeutic technology.

In summary, we have presented AIE polymersomes coated with a gold hemisphere that show increased autonomous propulsion via an AIE-mediated enhanced photothermal effect and that are capable of ROS production when irradiated using NIR. We have presented the complete synthetic approach for the preparation of TPEDC-rich PEG-PTMC$_{AIE}$ copolymers and assessed the impact of molecular composition upon self-assembly and characterized the fabrication of AIE/Au nanomo-tors using a range of physical methodologies. Functional testing of these bifunctional nanomotors confirmed their ability to undergo enhanced motion in proportion to the intensity of NIR irradiation, as provided by a two-photon laser. Nanomotors showed greater velocities than their mono-functional counterpart (without AIE), which indicated synergism between molecular

composition and motile behavior. Applying this promising system towards in vitro cellular testing, it was confirmed that upon NIR activation nanomotors showed an unmistakable and rapid onset of toxicity that was highly-localized to the region being irradiated. Taken together, this work outlines a uniquely functional nanomotor that pushes the boundaries of nanomedical engineering and photodynamic therapy.

## Methods

**Materials.** Poly(ethylene glycol) methyl ether (mPEG, $M_n$ 2 kDa) was purchased from Rapp Polymers. XPhos Pd G2 (Sigma), bis(pinacolato)diboron (99%, Sigma), potassium acetate (99%, Sigma), zinc powder (99%, Sigma), tetrakis(triphenyl-phosphine)palladium(0) (99%, Sigma), titanium(IV) chloride (99%, Sigma), 4-chloro-4-hydroxybenzophenone (98%, Sigma), bis(4-methoxyphenyl)methanone (98%, TCI), 4-bromobenzophenone (98%, Sigma), potassium carbonate ($K_2CO_3$, 99%, Sigma), trifluoromethanesulfonic acid (99%, Sigma), bis(pentafluorophenyl) carbonate (Apollo scientific, 97%), cesium fluoride (CsF, Sigma 99%), 2,2-bis (hydroxymethyl)propionic acid (Sigma, 99%), Cyanine7 NHS ester (Lumiprobe), Gold-Nanoshells (160 nm NanoXactTM Gold Nanoshells, nanoComposix), and all other chemicals were supplied by Sigma-Aldrich. Dialysis Membrane MWCO 12,000−14,000 Da from Spectra/Pro® was used for dialysis. Wheat Germ Agglu-tinin (WGA)-Alexa FluorTM 488 Conjugate, Hoechst 33342, LysoTrackerTM Green DND-26, propidium iodide (PI), Dulbecco's modified eagle medium (DMEM), phosphate-buffered salines (PBS, pH 7.4), no mycoplasma fetal bovine serum (FBS), trypsin-EDTA, and penicillin-streptomycin were obtained from Thermo-Fisher. All other chemicals were used as received without further treatment. All experiments conducted in this work use ultrapure Milli Q (Millipore) water (18.2 MΩ·cm).

**Preparation of AIE-polymersomes.** Taking PEG$_{44}$-P(AIE)$_5$ as an example, in a 4 mL vial, PEG$_{44}$-P(AIE)$_5$ (1 mg) was dissolved in 0.5 mL of THF and the vial was sealed with a rubber septum. The solution was stirred at 700 rpm for a minimum of 10 min prior to the addition of Milli-Q (0.5 mL, 0.25 mL h$^{-1}$) via a syringe pump. A needle was inserted into the septum to release pressure. The resulting cloudy suspension was transferred into a prehydrated dialysis bag (SpectraPor, MWCO: 12-14 kDa, 2 mL cm$^{-1}$). Dialysis was performed against Milli-Q water at room temperature for 24 h with a water change after 1 h. The physicochemical properties of AIE-polymersomes were characterized by dynamic light scatting, scanning electron microscopy, transmission electron microscopy, cryo transmission electron microscopy, and confocal laser scanning microscopy.

**Preparation of Janus AIE/Au nanomotors.** In order to construct AIE/Au nanomotors, a droplet of the AIE-polymersome solution (2 mg ml$^{-1}$) was dropped on a hydrophilic silica slide to form a monolayer of nanoparticles. After eva-poration in air, a turbo sputter coater (Quorum Technologies, K575X) was used to coat one side of the polymeric particles with a thin gold layer (65 mV, 30 s). Ultrasound treatment was used to re-disperse the Janus polymeric particles into aqueous solution. The size and morphology of AIE/Au nanomotors (AIE/Au NM) were characterized using dynamic light scatting, scanning electron microscopy, transmission electron microscopy and confocal laser scanning microscopy. EDX elemental mapping analysis of AIE/Au nanomotors was performed using SEM (Phenom ProX, The Netherlands).

**Preparation of cargo loaded AIE/Au nanomotors.** In order to test the integrity of the AIE/Au nanomotors, different cargoes (i.e., 10 kDa dextran-TMR and Cy7) loaded AIE-polymersomes were first prepared via a similar procedure compared with AIE-polymersomes; the dextran-TMR was dissolved in Milli-Q water and Cy7 was co-dissolved with the AIE-polymer. After dialysis purification, a droplet of the cargo-loaded AIE-polymersome solution (2 mg ml$^{-1}$) was dropped on a hydro-philic silica slide to form a monolayer of nanoparticles. After evaporation in air, a turbo sputter coater (Quorum Technologies, K575X) was used to coat one side of the polymeric particles with a thin gold layer (65 mV, 30 s). Ultrasound treatment was used to re-disperse the Janus polymeric particles into aqueous solution. The fluorescent emission behavior of cargo-loaded AIE/Au nanomotors (AIE/Au NM) was characterized using microplate reader.

**Integrity studies of Janus AIE/Au nanomotors.** The nanomotors integrity was checked by the leakage of the hydrophilic and hydrophobic cargoes. In detail, in the case of hydrophilic dextran-TMR loaded AIE/Au motors, the fresh prepared cargo loaded AIE/Au nanomotor (after ultrasound treatment and re-dispersed in aqu-eous medium) was divided into aliquots and incubated for different times. After a certain time (30 min, 1–5 h), the aliquots were centrifuged and the dextran-TMR emission intensity in the supernatant was measured using a micro-plate reader (with three replications) to determine the leakage of dextran-TMR cargo. In the case of hydrophobic Cy7 loaded AIE/Au motors, the freshly prepared cargo loaded AIE/Au nanomotors (after ultrasound treatment) were dispersed in aqueous solution. The emission intensity of Cy7 in this cargo-loaded AIE/Au nanomotor

was monitored by a microplate reader (with three replications) to determine the leakage of Cy7 cargo.

**Two-photon (TP) near-infrared (NIR)-activated motility.** The autonomous motion of AIE/Au nanomotors was observed and recorded by a TP-CLSM (Leica TCS SP5X) equipped with a ×40 water immersion microscope objective. AIE/Au nanomotors were detected by the intrinsic AIE fluorescent signal. Movement trajectories were tracked and analyzed by using Image J and Origin software. The TP NIR-infrared activated motility was determined according to previously published procedures. Based on the extracted trajectories, the velocity of NIR propelled AIE/Au nanomotors (V) was calculated following the formula: V = D/t after measuring both the traveled distance (D) and duration time (t). The diffusion coefficient (D) is defined as D = MSD/$i \cdot \Delta$t, where MSD is the mean square displacement (MSD), $\Delta$t is the time interval, and $i$ is the dimensional index. Here, for the case of two-dimensional analysis from the recorded videos, i is equal to 4. Corresponding mean square displacements (MSD) were then calculated following the reported equation: MSD = (x($\Delta$t)-x(0))$^2$ + (y($\Delta$t)-y(0))$^2$. The translational diffusion coefficient and rotational diffusion coefficient were calculated with the following Eq. (1):

$$D_T = \frac{k_B T}{6\pi\eta R} \quad \& \quad D_R = \tau_R^{-1} = \frac{k_B T}{8\pi\eta R^3} \tag{1}$$

(where $\eta$ is the viscosity and R the hydrodynamic radius of nanomotor). The theoretical values for particles with an average diameter ~400 nm are $D_T$ = 1.07 um$^2$s$^{-1}$, $D_R$ = 20 s$^{-1}$, $\tau_R$ = 0.05 s.

**Nanosight tracking analysis of motility.** Nanoparticle tracking analysis (NTA) was used to analyze the motion behavior of AIE/Au nanomotors, AIE polymersomes (AIE-Ps), and gold shells by using NanoSight S300. Samples were suspended in Milli-Q water to yield an approximate concentration of 10$^7$ and 10$^8$ particles per mL. For a typical experiment, 1 mL of sample (ca. 5 µg mL$^{-1}$) was loaded in the NTA chamber using a syringe. Then, the motion of AIE/Au nanomotors, AIE-polymersomes, and gold nanoshells was recorded for 30 s in triple. A 660 nm DPSS Red Diode Laser was utilized as external laser source to propel the particles. Different laser intensities were used during the experiments, including 0 W (laser off) and 1 W (laser on). The same measurement was performed 3 times to ensure reproducibility. The NTA 2.2 software allows the extraction and analysis of the trajectories of single particles. For each group, 30 nanoparticles were tracked for 30 s. Their mean squared displacements (MSD) were calculated following previously published procedures. MSD curves were extracted from the NTA recorded trajectories using the following Eq. (2):

$$MSD = [\Delta r^2(t)] = \left[ \frac{1}{N} \sum_{i=0}^{N} \left( r_i(t) - r_i(0) \right)^2 \right] \tag{2}$$

Where $r$ = radius and $t$ = sampling time and MSD (t) = 2dD, where $D$ = diffusion coefficient and $d$ = dimensionality (NTA measurements have dimension d = 2). The equation MSD = (4D)$\Delta$t + (v$^2$)($\Delta$t$^2$) was used to fit the MSD curves. From the fitting of the MSD curves, the average particle velocity was extracted. According to the particle diffusion coefficient, as described by Golestanian's diffusiophoretic model, a particle undergoing Brownian motion will display a linear MSD over time with the slope determined by the diffusion coefficient $D = K_B T/(6\pi\eta R)$. From this model, if the particles are in Brownian motion, the linear component of the MSD, according to the equation MSD = (4D)$\Delta$t, can be extracted. Indeed, in the absence of NIR light or 405 nm light irradiation, a linear relation between MSD and time was observed (Fig. 3e and h in main text, and Supplementary Fig. 44). The same linear relationship was also observed when control particles were exposed to light irradiation (Supplementary Figs. 45, 46, 48, and 49). In the presence of NIR light or 405 nm light, the gold shell coated asymmetric AIE/Au nanomotors displayed observable autonomous motion(In main text Fig. 3e and h, and Supplementary Fig. 50), the MSD curves (in the presence of light $\Delta$t > r) displayed a parabolic fit.

**Toxicity studies.** HeLa cells were obtained in TU/Eindhoven, and tested with mycoplasma contamination before use. HeLa cells were cultured in DMEM medium containing 10% FBS, 1% penicillin/streptomycin (complete DMEM) in 5% CO$_2$ at 37 °C. The relative cell viability was evaluated in vitro by an MTT assay. The cells were seeded in 96-well plates at a density of 5 × 10$^3$ cells per well in 100 µL complete DMEM medium and cultured for 24 h at 37 °C. Subsequently, the cells were incubated with the corresponding nanoparticles (AIE-polymersomes or AIE/Au nanomotors) at different concentrations for 24 h. The cells were washed and fresh medium containing MTT was added into each plate. The cells were incubated for another 4 h. After removing the medium containing MTT, dimethyl sulfoxide (100 µL) was added to each well to dissolve the formazan crystals. Finally, the plate was gently shaken for 5 min and the absorbance at 490 nm was recorded with a micro-plate reader.

**Cell membrane disruption with AIE/Au nanomotors.** HeLa cells were cultured in DMEM medium containing 10% FBS, 1% penicillin/streptomycin (complete DMEM) in 5% CO$_2$ at 37 °C. HeLa cells were seeded in a µ-slide eight

well plate for 24 h, and then the medium was refreshed. The cells were stained with wheat germ agglutinin Alexa Fluor-TM 488 conjugate and Hoechst 33342 to show the cell membrane, as well as with propidium iodide (PI) to show the real-time enhanced permeability of cell membranes. The cells were washed with PBS twice. Thereafter, the cells were treated with AIE/Au nanomotors (25 µg ml$^{-1}$). Immediately, cells were subjected to a two-photon confocal NIR laser and the fluorescence images of the cells were captured using a Leica TCS 264 SP5X system.

**ROS production of AIE/Au nanomotors within cells.** HeLa cells were cultured in DMEM medium containing 10% FBS, 1% penicillin/streptomycin (complete DMEM) in 5% CO$_2$ at 37 °C. HeLa cells were seeded in a µ-slide eight well plate for 24 h, and then the medium was refreshed. The cells were subsequently loaded with CM-H2DCF for 0.5 h and stained with Hoechst 33342 for 10 min. Then the cells were washed and treated with AIE/Au nanomotors (25 µg ml$^{-1}$). Immediately, the cells were subjected to a two-photon confocal NIR laser and the fluorescence images of the cells were captured using a Leica TCS 264 SP5X system.

**Cell necrosis induced by AIE/Au nanomotors.** HeLa cells were cultured in DMEM medium containing 10% FBS, 1% penicillin/streptomycin (complete DMEM) in 5% CO$_2$ at 37 °C. HeLa cells were seeded in a µ-slide 8 well plate for 24 h, and then the medium was refreshed. Then, the cells were stained with Hoechst 33342 for 10 min. Thereafter, the cells were washed and incubated with calcein for live-cell staining, and PI for dead cell staining for 10 min. Next, the cells were treated with AIE/Au nanomotors (25 µg ml$^{-1}$). Immediately, the cells were subjected to a two-photon confocal NIR laser and the fluorescence images of the cells were captured using a Leica TCS 264 SP5X system.

**Reporting summary.** Further information on research design is available in the Nature Research Reporting Summary linked to this article.

## Data availability
The data that support the findings of this study are available from the corresponding author upon reasonable request. Source data are provided with this paper.

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

## Acknowledgements

The authors would like to acknowledge the ERC Advanced Grant Artisym 694120, the Dutch Ministry of Education, Culture and Science (Gravitation program 024.001.035), the NWO-NSFC Advanced Materials (project 792.001.015), and the European Union's Horizon 2020 research and innovation program Marie Sklodowska-Curie Innovative Training Networks Nanomed, (No. 676137) for funding. We thank the Ser Cymru II program for support of DSW; this project received funding from the European Union's Horizon 2020 research and innovation program under the Marie Skłodowska–Curie grant agreement No. 663830.

## Author contributions

S.C. and J.S. contributed equally to this work. S.C., J.S, H.W. L.K.E.A.A., D.S.W., and J.C.M.H. wrote the manuscript; S.C., J.S., H.F., L.K.E.A.A., D.S.W., and J.C.M.H. designed the research; S.C., M.T.D.M., J.S., H.W. S.S., T.M. and I.A.B.P performed the experiments. All authors reviewed the manuscript.

## Competing interests

The authors declare no competing interests.
