## [Peer Review File · Nature Communications]

Reviewers' comments:

Reviewer #1 (Remarks to the Author):

In this manuscript, the authors present phototactic/phototherapeutic nanomotors by combining ROS-generating AIE polymersomes and asymmetrically coated Au nanoshells. The approach is appealing for the following reasons. On one hand biodegradable block copolymers decorated with AIE motifs can harness two photon near-infrared (TP-NIR) radiation and emit light absorbable by the asymmetric Au nanoshell. This additional absorption reinforces the directional thermophoretic motion of the Janus nanoparticles, which is firstly actuated by the direct absorption of TP-NIR radiation by Au nanoshell as described in a previous work of the same group (ACS Nano 12, 4877-4885 (2018)) and by other groups earlier (Phys. Rev. Lett. 105, 268302 (2010), J. Am. Chem. Soc. 138, 6492-6497 (2016)). On the other hand, the hybrid nanomotors can simultaneously produce ROS for phototherapeutic purpose because of the presence of AIEgens. Both features have been illustrated in this work by a series of physical studies and in vitro tests.

The use of AIEgens has been appropriately justified and well characterized. However, the use of the vesicular structures (polymersomes) is questionable. The charm of polymersomes is their big aqueous compartment enclosed by a thick polymer bilayer (compared to thin lipidic bilayer). Polymersomes can be used to encapsulate both hydrophilic and hydrophobic substances. But here with the Au sputtering procedure, the polymersomes are normally broken or crashed, as visibly shown in Fig. 2d and 2e. This is the case for most of polymersomes, maybe except for very small ones ($D < 50$ nm), or multilamellar ones, or those with rigid and robust membrane. The polymersomes here, with diameter $D = 300 - 500$ nm and thickness of 7.7 nm, do not belong to these exceptions. By the way, how did the authors deduce the molecular interdigitated organization in the 7.7nm-thick membrane shown in Figure S25? To prove the sketched Janus polymersomes in all figures Fig. 1-4, cryo-EM is necessary and required after their redispersion in aqueous medium. This reviewer does not believe the schematic presentation of the intact round Janus vesicles.

Reviewer #2 (Remarks to the Author):

The authors present a description of the fabrication process of self-propelled particles which are functionalized with AIEgenic molecules. When such molecules are irradiated with light, they produce reactive oxygen species (ROS) which is used in phototherapy to e.g. induce cell death or to reduce cell activity. The particles are fabricated from biodegradable AIE polymers which are coated by an asymmetric gold nano shell which results in a functionalized nanoscopic Janus particle. When illuminated with NIR light, plasmons are excited within the Au cap which causes cap heating and eventually a thermophoretic self-propulsion. The authors also demonstrate, that the presence of AIE enhances self-propulsion due to transducing radiant energy into Au-shell which is confirmed by the strong wavelength-dependence of the illuminating light. They also found evidence for directional motion of such nanomotors away from the light source which is attributed to negative phototaxis. Finally, they applied their system to a number of cell assays from which they find an enhanced cytotoxicity of their system when activated with light.

I found this manuscript rather difficult to read, perhaps the authors tried to squeeze too much information into it. This is at the expense of clarity in the presentation. In the current state, I do not recommend this paper to be published in Nat. Comm.

For example, the characterization of the self-propulsion should be very much expanded. Apart from a few trajectories and MSDs, only little information is found in the paper. How does the measured

hydrodynamic size of their particles compares to the crossover-time from ballistic to effective diffusive behavior in their Figs.3c,d ? Does the long-time effective diffusion coefficient compares to the rotational diffusion? I was expecting such type of cross-checks to confirm the validity of their system.

Also, I am not sure about their interpretation of negative phototaxis. Phototaxis typically requires a light gradient (see e.g. their Ref 28.). Where this that light gradient coming from? How large is that gradient? I was also missing a clear motivation, why phototaxis is important in the context of the suggested medical applications.

I would expect that nanomotors with negative phototactic behavior should rapidly leave the illuminated region. This, however, is not what the authors want because then the phototherapeutic function is lost, right?

Why activated nanomotors have a higher toxicity than non-irradiated ones? In other words, what is the connection between particle motility and toxicity?

Reviewer #3 (Remarks to the Author):

The authors are presenting a polymersome based nanomotor that shows AIEgenic fluorescent properties.

Their detailed knowledge on the synthetic approach for fabricating AIEgenic polymersomes is given in the SI, which is very comprehensive and well done.

I have a few doubts when it comes to the motility studies. Even though the authors show in-vitro studies at the end, none of their lab studies gives information about the influence of salt contents etc. The trackings are done in MilliQ water and do not take into account that in real environments the nanomotors have to swim in high ionic strength media. The gap between these two scenarios is very big, so I find it difficult to connect them. It would help to see the motion in a normal microscopy experiment in the biological medium.

When it comes to the motion studies, the videos that the authors present are very badly labelled. Since there are no big visible differences between the individual cases, it would help the reader if the videos were named accordingly or the caption could be incorporated directly into the video.

Additionally, I would suggest to graft AuNP on the polymersomes instead of using AuShells, that would reveal a more distinct influence of the presence of gold and give a clear hint on the origin of the synergy between the propulsion and the phototherapy.

A curiosity, that probably goes beyond the scope of the manuscript is the origin of the phototactic behaviour of these micromotors.

Reviewers' comments:

Reviewer #1 (Remarks to the Author):

In this manuscript, the authors present phototactic/phototherapeutic nanomotors by combining ROS-generating AIE polymersomes and asymmetrically coated Au nanoshells. The approach is appealing for the following reasons. On one hand biodegradable block copolymers decorated with AIE motifs can harness two photon near-infrared (TP-NIR) radiation and emit light absorbable by the asymmetric Au nanoshell. This additional absorption reinforces the directional thermophoretic motion of the Janus nanoparticles, which is firstly actuated by the direct absorption of TP-NIR radiation by Au nanoshell as described in a previous work of the same group (ACS Nano 12, 4877-4885 (2018)) and by other groups earlier (Phys. Rev. Lett. 105, 268302 (2010), J. Am. Chem. Soc. 138, 6492-6497 (2016)). On the other hand, the hybrid nanomotors can simultaneously produce ROS for phototherapeutic purpose because of the presence of AIEgens. Both features have been illustrated in this work by a series of physical studies and in vitro tests.

We are pleased with the positive evaluation of the reviewer and we have endeavored to address their comments in a systematic fashion.

The use of AIEgens has been appropriately justified and well characterized. However, the use of vesicular structures (polymersomes) is questionable. The charm of polymersomes is their big aqueous compartment enclosed by a thick polymer bilayer (compared to thin lipidic bilayer).

As identified in our manuscript, the design rationale behind our system is related to: (i) relatively small size in comparison to micromotor technology as published by various authors and (ii) to achieve large enough particle dimensions to enable sputter coating for Au Janus structures. Polymersomes, as opposed to small micelles, fulfil this target ambition perfectly and can enable future applications through cargo encapsulation.

Polymersomes can be used to encapsulate both hydrophilic and hydrophobic substances. But here with the Au sputtering procedure, the polymersomes are normally broken or crashed, as visibly shown in Fig. 2d and 2e. This is the case for most of the polymersomes, maybe except for very small ones ($D < 50$ nm), or multilamellar ones, or those with rigid and robust membrane. The polymersomes here, with diameter $D = 300 - 500$ nm and thickness of 7.7 nm, do not belong to these exceptions.

It should be noted that figure 2a and 2e present 'dry' techniques whereby particles are dried prior to imaging – this causes polymersomal structures such as those comprising thin membranes (as in this instance) to collapse. This is evident in figure 2a and figures S26-S27 and in the updated S31a – under dry conditions one cannot judge the stability of particles in relation to their spherical nature as they all typically appear as deflated footballs. In fact, from 2d we see that the structures look rather more robust as compared to the uncoated counterpart. Cryo-TEM images (eg. figure 2b) show that the deflated spheres present in figure 2a are, in fact, fully inflated when captured in their native (wet) state.

Any deflation of the particles after coating should be evident in our DLS data. As presented in figure S30, no such deflation is observed and only a small increase in hydrodynamic size is detected after gold coating.

Repeat TEM images of AIE/Au nanomotors are provided in figure S32 to demonstrate the uniformity of re-dispersed nanomotors.

To go beyond this, and to provide confidence in the preservation of an essential property of polymersomes after gold coating (i.e. the encapsulation of hydrophobic and hydrophilic cargoes) we conducted additional experiments where fluorescent dextran-TMR and (in a separate experiment) Cy7 were loaded into the AIE/Au nanomotor. This data is provided in figure S35 and highlighted in text on page 6.

By the way, how did the authors deduce the molecular interdigitated organization in the 7.7nm-thick membrane shown in Figure S25?

The structure presented in Figure S25 was a schematic of polymersome self-assembly that is in line with a number of reviews (Ref: Eisenberg, et. al., Chem. Soc. Rev. 2012, 41, 5969-5985; Landfester, et. al., Chem. Soc. Rev. 2018, 47, 8572-8610), however, we have altered this so as to prevent inadvertent misrepresentation of molecular configuration so as to avoid any possible confusion.

To prove the sketched Janus polymersomes in all figures Fig. 1-4, cryo-EM is necessary and required after their redispersion in an aqueous medium. This reviewer does not believe the schematic presentation of the intact round Janus vesicles.

As the gold coating was identified and visible from TEM and SEM images (including EDX analysis) in figures 2 and S31-S33 we were confident about the stability of our particles (also confirmed by DLS). The reviewers' lack of belief in this fact seems to stem from a presupposition that the dry techniques used should give spherical particles – this is not the case with such 'hollow' polymersome nanostructures as evidenced by comparisons between figure 2a and 2b.

To stem the concerns of the reviewer we have produced additional dry TEM images (figure S32), cryo-TEM image (figure S34) and DLS data (figure S30) to support particle stability.

Reviewer #2 (Remarks to the Author):

The authors present a description of the fabrication process of self-propelled particles which are functionalized with AIEgenic molecules. When such molecules are irradiated with light, they produce reactive oxygen species (ROS) which is used in phototherapy to e.g. induce cell death or to reduce cell activity. The particles are fabricated from biodegradable AIE polymers which are coated by an asymmetric gold nano shell which results in a functionalized nanoscopic Janus particle. When illuminated with NIR light, plasmons are excited within the Au cap which causes cap heating and eventually a thermophoretic self-propulsion. The authors also demonstrate, that the presence of AIE enhances self-propulsion due to transducing radiant energy into Au-shell which is confirmed by the strong wavelength-dependence of the illuminating light. They also found evidence for directional motion of such nanomotors away from the light source which is attributed to negative phototaxis. Finally, they applied their system to a number of cell assays from which they find an enhanced cytotoxicity of their system when activated with light.

I found this manuscript rather difficult to read, perhaps the authors tried to squeeze too much information into it. This is at the expense of clarity in the presentation. In the current state, I do not recommend this paper to be published in Nat. Comm.

We appreciate that the reviewer has compiled a concise overview of the manuscript, which combines multiple elements in a compact form including chemical synthesis, functional testing and in vitro cell assays. Such a research communication is, of course, challenging to write and refers to many areas of expertise. We have used the comments below to make changes to the manuscript and hope that these provide greater clarity to the text.

For example, the characterization of the self-propulsion should be very much expanded. Apart from a few trajectories and MSDs, only little information is found in the paper. How does the measured hydrodynamic size of their particles compares to the crossover-time from ballistic to effective diffusive behavior in their Figs.3c,d ? Does the long-time effective diffusion coefficient compares to the rotational diffusion? I was expecting such type of cross-checks to confirm the validity of their system.

Indeed, we agree with the reviewer that a robust understanding of the motile behavior and properties of this nanomotor system is important. To clarify, in its current state our manuscript has emphasized:

1. The relation between laser power and increasing diffusion velocity
2. The relation between laser wavelength and the AIE/Au modules of our nanomotor
3. That non-Brownian motion is clear

The next-level discussion relating to ballistic, directional motion was initially muted in our manuscript due to the need for extreme caution when making claims related to such mechanisms when the underpinning theory is so poorly understood for nanomotors. With the comments from the reviewer we have now undertaken additional experimental work and our revised manuscript now provides a more

detailed discussion and expands upon our previous observations with more attention given to transitions between diffusive and ballistic propulsion in our hybrid nanomotors.

To start with (answering the reviewers' specific comments), when considering nanomotors with an average hydrodynamic diameter of ~ 400nm, according to equations:

$$D = \frac{k_B T}{6 \pi \eta R} \quad \& \quad \tau_R = \frac{R^2}{D}$$

(where η is the viscosity and R the hydrodynamic radius of nanomotor)

The theoretical values are $D_T = 1.07 \text{ um}^2\text{s}^{-1}$, $D_R = 20 \text{ s}^{-1}$, $\tau_R = 0.05\text{s}$.

As the time between frames using either our confocal or nanosight apparatus is about 0.5s or 0.03s, respectively, we should, according to current theory, not capture the ballistic regime. However, as observed in our videos (supplementary video S3, S5 and S7) and subsequent analysis, there is clearly directional (ballistic) motion observed in our motors.

We have now given more attention to the discussion surrounding this interesting observation and have expanded our text to discuss how our hybrid (AIE/Au) nanomotors display ballistic motion in response to increasing laser intensity - overcoming theoretical limitations. Overall, we attribute this effect to enhanced thermophoretic mobility arising from local heating of the Au nanoshell, an effect that is well documented in the literature and is known to display non-linear response both in terms of laser power and duration of irradiation (Refs: Nam, et al., *Adv. Sci.* 2019, 6, 1900471; He, et al., *J. Am. Chem. Soc.* 2016, 138, 6492–6497; He, et al., *Angew. Chem. Int. Ed.* 2018, 57, 12463 –12467).

As a cross-check, analysis of trajectories (from confocal microscopy) without laser power yielded D_T values that were very close to theory (AIE/Au hybrids = 0.85 and Au-only control = 0.93). This is now included in the text on page 6/7.

In our data with increasing laser power there is a clear transition from diffusive to directional/ballistic/propulsive motion. Looking into the literature regarding the plasmonic heating of Au NPs, Chen and He *et al* show the non-linear trend in *bulk* heating in response to NIR irradiation (Ref: Chen, et al., *Acc. Chem. Res.* 2015, 48, 2506–2515; Chen, et al., *Angew. Chem. Int. Ed.* 2013, 52, 13958 – 13964; He, et al., *J. Am. Chem. Soc.* 2016, 138, 6492–6497). We hypothesize that this would correlate to local heating that would drive a non-linear response in terms of thermophoresis; increased in the case of our AIE-enhanced nanomotors but still evident for control (Au only) nanomotors; showing ballistic motion at timeframes up to 10 seconds. As to *why* an enhanced thermophoretic effect arising from local heating of the Au nanoshell results in directional (ballistic) motion – we hypothesize that this would have a significant effect on τ_R . If the propulsion driven by local (asymmetric) heating overpowers 'normal' τ_R and creates a thermal wave or trajectory along which particles will tend to propel then this could account for this observation, however, such speculation is very difficult to articulate within such a manuscript that is already loaded with experimental data.

We have included additional discussion on pages 6-8 and highlighted the non-linear trend in propulsion at time scales up to 10 seconds, giving explanation for this in line with the thermal effect of plasmonic gold but stopping short of broad speculation regarding a precise mechanism for this.

It should then be noted that our data collected from the nanosight is up to 0.51 s and clearly shows ballistic behavior. This can be explained by the role of laser power in driving enhanced propulsion. Higher laser powers would achieve more rapid onset of local (plasmonic) heating and, therefore, give rise to propulsive motile behavior.

Also, I am not sure about their interpretation of negative phototaxis. Phototaxis typically requires a light gradient (see e.g. their Ref 28.). Where this that light gradient coming from? How large is that gradient? I was also missing a clear motivation, why phototaxis is important in the context of the suggested medical applications.

In the literature, phototaxis can be achieved by controlled directional light irradiation (Tang, et al., Nature Nanotechnology volume 11, pages 1087-1092, 2016) – with our focus upon potential therapeutic application we were keen to explore this effect and gain insight into the directionality of our motors.

Using Nanoparticle Tracking Analysis (NanoSight) in combination with a laser source, we activated and monitored the motion of the hybrid AIE/Au nanomotors. The experimental setup was shown in Figure 3f, where the laser pathway passes through the sample in a fixed manner. In terms of a light gradient, under normal circumstances we might expect laser intensity to drop off with $1/r^2$ (where r is the distance from illumination source), however, given the colloidal nature of our solution this will be significantly increased due to Rayleigh scattering – giving rise to a steep intensity gradient. However, the magnitude of this gradient remains elusive suffice to say that it is much greater than $1/r^2$.

According to the motion video (video S5) and normalized trajectory (Figure 3g and Figure S44), the hybrid AIE/Au nanomotors exhibit directional motion and propel away from the light source. Consequently, the nanomotors designed in this work experience negative phototaxis.

Such phototactic behavior opens the possibility of optical navigation as a feasible method to achieve high-precision manipulation of nanomotors in vivo.

Additional discussion has been added on page 8 to highlight this aspect of the study.

I would expect that nanomotors with negative phototactic behavior should rapidly leave the illuminated region. This, however, is not what the authors want because then the phototherapeutic function is lost, right?

Under a vertically focused confocal TP-NIR laser, nanomotors with a negative photo-tactic behavior displayed motion away from the laser source, as confirmed by confocal trajectory tracking (Figure 3b). With such active mobility, when compared with the static control motors (video S12), our active nanomotor showed enhanced interaction with cancer cells (Figure S53 and video S13). In our system, it appears that NIR irradiation can be used to excite nanomotors and drive them in the forward direction (away from the laser source), which is a form of negative phototaxis that is very useful in therapeutic application (Refs: Pumera, et al., Chem. Soc. Rev., 2019, 48, 4966–4978; He, et al., Angew. Chem. Int. Ed. 2018, 57, 12463 –12467; Yang, et al., Nat. Photon. 2015, 9, 563-571; Xu, et al., Applied Materials Today 2020, 18, 100504; Lohmüller, et al., Nano Lett. 2018, 18, 7935-7941).

Why activated nanomotors have a higher toxicity than non-irradiated ones? In other words, what is the connection between particle motility and toxicity?

Our strategy was two-fold. Firstly, to use propulsion as means to enhance cellular interactions and undermine the stability of the plasma membrane (akin to cell percolation as reported by He, *et al.*) (Ref: *Angew. Chem. Int. Ed.* 2018, 57, 12463 –12467). Secondly, to generate reactive oxygen species (ROS) locally in order to toxify the cells and induce highly localized cell killing as a therapeutic modality commonly implemented for radiotherapy (e.g. PDT). Thirdly, there is a chance that localized heating could also contribute to this via a photothermal effect.

Combining these elements has never been demonstrated in such a system towards a highly efficient (and precise) nanomedicine that can be activated using TP-NIR. Furthermore, our system is of smaller size as compared to the numerous micromotor examples – giving much greater potential impact for therapeutic application.

Additional discussion has been added on page 8 to highlight the transition from nanomotor to therapeutic application.

Reviewer #3 (Remarks to the Author):

The authors are presenting a polymersome based nanomotor that shows AIEgenic fluorescent properties.

Their detailed knowledge on the synthetic approach for fabricating AIEgenic polymersomes is given in the SI, which is very comprehensive and well done.

I have a few doubts when it comes to the motility studies. Even though the authors show in-vitro studies at the end, none of their lab studies gives information about the influence of salt contents etc. The trackings are done in MilliQ water and do not take into account that in real environments the nanomotors have to swim in high ionic strength media. The gap between these two scenarios is very big, so I find it difficult to connect them. It would help to see the motion in a normal microscopy experiment in the biological medium.

We thank the reviewer for their insightful comments.

Experiments using TP-NIR laser-induced motion, recorded with confocal microscopy, were performed in PBS medium. Real-time interactions with cells and therapeutic studies were also conducted in PBS medium to ensure comparable results obtained between experiments.

Although initial NTA studies were conducted in Milli-Q water, we have repeated these measurements to show the same NIR-activated motility is observed in cell culture medium as in Milli-Q water (Figure S45, and video S6-S7).

When it comes to the motion studies, the videos that the authors present are very badly labeled. Since there are no big visible differences between the individual cases, it would help the reader if the videos were named accordingly or the caption could be incorporated directly into the video.

We have updated the information and names of the videos, with a list found in the SI on page 1.

Additionally, I would suggest to graft AuNP on the polymersomes instead of using AuShells that would reveal a more distinct influence of the presence of gold and give a clear hint on the origin of the synergy between the propulsion and the phototherapy.

Spherical AuNPs do not possess the same absorbance properties as those of nanoscopic Au shells and rarely exceed 650nm, giving poor photothermal properties (Ref: Nordlander, et. al., Nature Nanotechnology 10, 25–34, 2015; Nam, et. al., Adv. Sci. 2019, 6, 1900471; Xia, et. al., Chem. Rev. 2015, 115, 10410–10488). Gold nanoshells are a recognized pathway to achieve such Janus structures with excellent photothermal properties and potential propulsion via self-thermophoresis (Ref: Sano, et. al., Phys. Rev. Lett. 105, 268302, 2010; Pumera et. al., Chem. Soc. Rev., 2019, 48, 4966–4978).

In our studies we use extensive controls to ensure that robust conclusions can be drawn from our studies and we demonstrate that either component (AIE or Au nanoshell) on their own do not perform anywhere near as well as the hybrid nanomotor system both in terms of propulsion and towards localized cell ablation.

A curiosity that probably goes beyond the scope of the manuscript is the origin of the phototactic behavior of these micromotors.

Our nanomotors show negative phototactic behavior, moving away from the laser light source. This property has been observed in the literature and can be attributed to the movement of motors in line with the decreasing gradient of transmitted irradiation (i.e. away from the laser source) (Ref: Pumera, et al., *Chem. Soc. Rev.*, 2019, 48, 4966—4978; Bechinger, et al., *Nature Communications* volume 7, 12828, 2016; Tang, et al., *Nature Nanotechnology* volume 11, pages 1087–1092, 2016).

Reviewers' comments:

Reviewer #1 (Remarks to the Author):

Reply to rebuttal (reviewer #1):

Regarding to the authors' reply, this reviewer considered that the main questions have not been addressed.

Polymersomes can be used to encapsulate both hydrophilic and hydrophobic substances. But here in the Au sputtering procedure, dry sample and high vacuum are necessary. Therefore, the polymersomes with diameter $D = 300 - 500$ nm and thickness of 7.7 nm are broken or crashed, as visibly shown in Fig. 2d and 2e.

"Figure 2a and 2e present 'dry' techniques whereby particles are dried prior to imaging – this causes polymersomal structures such as those comprising thin membranes (as in this instance) to collapse." Sure. But the Au sputtering procedure also need the particles to be dried previously. Even they are not dried completely, they will be in the sputter. So, this reviewer does not think the polymersomes can be kept intact after Au sputtering.

Cryo-TEM image of Janus AIE/Au nanomotors after re-dispersion into Milli-Q water was performed by authors (Figure S34). If we compare carefully this image with the cryo-EM image of polymersomes before Au sputtering (Figure S25), we can notice that the membrane structure (dark periphery giving the thickness measurement, which was caused by membrane return in the edge of spherical vesicle) is absent, especially in the left sides of particles without Au layer. This reviewer is still not convinced of the schematic presentation of the intact round Janus vesicles in all figures Fig. 1-4.

TEM and SEM images added by authors (S31-S32) still cannot prove the stability of the vesicular structures after Au sputtering, because they are all dry techniques. Supplemental DLS data S30 is the same, because the maintain of hydrodynamic size cannot prove the vesicular structure.

"To go beyond this, and to provide confidence in the preservation of an essential property of polymersomes after gold coating (i.e. the encapsulation of hydrophobic and hydrophilic cargoes) we conducted additional experiments where fluorescent dextran-TMR and (in a separate experiment) Cy7 were loaded into the AIE/Au nanomotor. This data is provided in figure S35 and highlighted in text on page 6."

This reviewer has some questions about the choice of cargoes. Firstly, hydrophilic fluorescent dextran-TMR is a macromolecule of 10 KDa. If the polymersome is intact, this macromolecule should not be release at all, while a release of 10% was observed after only 2h. Could the dextran-TMR macromolecules go out through the cracked places in collapsed vesicles? As for the hydrophobic Cy7 encapsulated in the hydrophobic part of the membrane, it should not be significantly released even with vesicle breaking or crashing.

Reviewer #2 (Remarks to the Author):

Again, the authors have answered some of my questions. However, I have to admit, that – despite their wordy answers – I am not truly satisfied. The role of a light gradient for phototaxis is still not clear to me even after reading their answer to my related question.

So, my original assessment has not really changed. My feeling is, that the interpretation of the data is not always fully transparent, however, my view may be slightly biased from a physicists perspective. In case, referee 1 is in full support of this manuscript, I am happy to join.

Reviewer #1 (Remarks to the Author):

Regarding to the authors' reply, this reviewer considered that the main questions have not been addressed.

Polymersomes can be used to encapsulate both hydrophilic and hydrophobic substances. But here in the Au sputtering procedure, dry sample and high vacuum are necessary. Therefore, the polymersomes with diameter $D = 300 - 500$ nm and thickness of 7.7 nm are broken or crashed, as visibly shown in Fig. 2d and 2e.

It should be noted that on page 5, line 122-125, we clearly clarify that the Janus nanomotors were prepared from PEG₄₄-P(AIE)₁₄, which has a thick and robust membrane thickness around 14 nm. The images from previous figure 2 (panels d and e) were collected using 'dry' electron microscopy techniques where the samples have been imaged in their non-hydrated state – as such this is not representative of their solution structure. However, we recognize the critique and have undertaken to explore the structure of our nanoparticles in a 'hydrated' state using cryo-TEM, which captures the native solution-state structure of the particles.

"Figure 2a and 2e present 'dry' techniques whereby particles are dried prior to imaging – this causes polymersomal structures such as those comprising thin membranes (as in this instance) to collapse." Sure. But the Au sputtering procedure also need the particles to be dried previously. Even they are not dried completely, they will be in the sputter. So, this reviewer does not think the polymersomes can be kept intact after Au sputtering.

To confirm the integrity of the polymersomes after drying, we conducted additional control experiments in which dried polymersomes were re-dispersed in MQ water without Au sputtering. In the newly added Figure S34, the drying process (equivalent to that used in the sputtering process) and re-dispersion did not significantly affect polymersomal morphology and vesicular nature; confirmed by DLS and cryo-TEM imaging.

The sputtering coating methodology used in this work was found to be compatible with polymeric vesicles with sizes ranging from 100-300 nm (Ref: Adv. Funct. Mater. 2018, 1706117). In our contribution, in page 5, line 122-125, we stated that the polymersomes used in the sputter coating are ~300 nm

polymersomes with a membrane thickness of 14 nm (Table S2), which is robust and able to withstand such processing without irreversible structural collapse (in line with the literature).

To further check the integrity and vesicular nature of the Janus nanomotors (after Au coating), cryo-electron tomography (cryo-ET) was performed. In the newly updated Figure 2e (where the high resolution cryo-TEM image replaces the previous dry TEM image to highlight the 'hydrated' state) and Figure S35, the membrane structure is clearly visible in high resolution cryo-TEM (with higher defocus value applied), which is more pronounced in the inverse contrast image (background subtracted), accompanied by the very high Au contrast. Furthermore, high resolution cryo-TEM at different tilting angles (Figure S37) further proved the nearly spherical structure after sputter coating and subsequent hydration. As further evidence, our cryo-ET 3D reconstruction results confirmed (Figure S36, supplemental video S1-2) that the Au coating is isolated to the surface of polymersomes, highlighting their integrity after Au coating.

Cryo-TEM image of Janus AIE/Au nanomotors after re-dispersion into Milli-Q water was performed by authors (Figure S34). If we compare carefully this image with the cryo-EM image of polymersomes before Au sputtering (Figure S25), we can notice that the membrane structure (dark periphery giving the thickness measurement, which was caused by membrane return in the edge of spherical vesicle) is absent, especially in the left sides of particles without Au layer. This reviewer is still not convinced of the schematic presentation of the intact round Janus vesicles in all figures Fig. 1-4.

To further provide proof to support our findings, we repeated high resolution cryo-TEM focusing on the contrast between polymersome membrane and gold. As displayed in newly added high resolution cryo-TEM (with higher defocus value applied) (Figure S35-S37), membrane structure and gold coating are clearly distinguishable in the inverse contrast image (background subtracted) of the Janus-polymersomes. Cryo-TEM imaging at different tilting angles and 3D tomographic reconstruction (Figure S36-37, supplemental video S1-2) further confirmed the round and vesicular nature of the intact polymersomes with gold coating.

TEM and SEM images added by authors (S31-S32) still cannot prove the stability of the vesicular structures after Au sputtering, because they are all dry techniques. Supplemental DLS data S30 is the same, because the maintain of hydrodynamic size cannot prove the vesicular structure.

To address this reviewer's concern, we again repeated high resolution cryo-TEM, cryo-TEM at different tilting angles and cryo-ET to further support our finding (in the above response).

"To go beyond this, and to provide confidence in the preservation of an essential property of polymersomes after gold coating (i.e. the encapsulation of hydrophobic and hydrophilic cargoes) we conducted additional experiments where fluorescent dextran-TMR and (in a separate experiment) Cy7 were loaded into the AIE/Au nanomotor. This data is provided in figure S35 and highlighted in text on page 6."

This reviewer has some questions about the choice of cargoes. Firstly, hydrophilic fluorescent dextran-TMR is a macromolecule of 10 KDa. If the polymersome is intact, this macromolecule should not be released at all, while a release of 10% was observed after only 2h. Could the dextran-TMR macromolecules go out through the cracked places in collapsed vesicles? As for the hydrophobic Cy7

encapsulated in the hydrophobic part of the membrane, it should not be significantly released even with vesicle breaking or crashing.

The fabrication of Janus-nanoparticles includes drying of the polymersomes, sputter coating and sonication processes. These procedures will cause slight defects in the polymersome that, we believe, contributed to the initial release of 10% of encapsulated cargo. However, cargo leakage did not increase with prolonged incubation times (5 h), which indicated that at least the vast majority (90%) of the Janus-polymersome particles are intact.

REVIEWERS' COMMENTS

Reviewer #1 (Remarks to the Author):

The additional analysis with cryo-EM and cryo-ET have shown effectively that the nanomotors are intact vesicular systems.

Reviewer comments:

Reviewer #1 (Remarks to the Author):

The additional analysis with cryo-EM and cryo-ET have shown effectively that the nanomotors are intact vesicular systems.

We thank the reviewer for their assessment